# A Parallel Multi-compartment Spiking Neuron For Multi-scale Sequential Modeling

## Abstract

The human brain possesses remarkable abilities in processing sensory signals that exhibit complex temporal dynamics. However, brain-inspired Spiking Neural Networks (SNNs) encounter challenges when dealing with sensory signals that have a high temporal complexity. These challenges primarily arise from the utilization of simplified spiking neuron models, such as the widely adopted Leaky Integrate-and-Fire (LIF) model, which has limited capability to process temporal information across multiple time scales. Additionally, these spiking neuron models can only be updated sequentially in time, resulting in slow training processes that pose particular difficulties when dealing with long sequences. To address these issues, we propose a novel Parallel Multi-compartment Spiking Neuron (PMSN), inspired by the multi-compartment models that simulate the temporal dynamics of biological neurons involved in sensory processing and memory. The PMSN model captures the intricate interactions among various neuronal compartments, allowing multi-scale temporal information to be preserved and integrated for effective sequential modeling. Furthermore, the PMSN model is meticulously designed to facilitate parallel training on GPU-accelerated machine learning frameworks. Our experimental results across numerous sequential modeling tasks demonstrate the superior performance of the proposed PMSN model compared with other spiking neuron models. Specifically, it exhibits enhanced accuracy, accelerated simulation, and favorable trade-offs between accuracy and computation cost.

## 1 Introduction

The human brain, recognized as one of the most sophisticated computational systems on the planet, exhibits remarkable performance in processing a wide range of sensory signals. Spiking Neural Networks (SNNs), designed to emulate the fundamental structure and operational principles of the human brain, hold great potential to replicate the exceptional sequential modeling capacity observed in the human brain. Recent advancements in training algorithms (Wu et al., 2018; Guo et al., 2022; Meng et al., 2022; Zhang et al., 2022) and neural architectures (Jeffares et al., 2022; Yao et al., 2022; Sengupta et al., 2019; Fang et al., 2021a) have substantially enhanced the capabilities of SNNs. Notably, for tasks that require limited temporal context, such as image classification, SNNs demonstrate competitive performance while achieving substantial energy savings compared to traditional Artificial Neural Networks (ANNs) (Bu et al., 2022; Yang et al., 2022; Wu et al., 2021a;b).

However, due to the inherent memory constraints, the simplified spiking neurons employed in previous research encounter significant challenges when confronted with tasks involving complex temporal dynamics. For instance, the speech recognition task requires the model to preserve and integrate information across multiple time scales, spanning from phonemes, words, to sentences, a capacity we call **multi-scale sequential modeling**. To address this issue, recent studies have introduced novel neuron models that incorporate adaptive variables (Bellec et al., 2018; Yin et al., 2021; Fang et al., 2021b) to establish temporal dependency across distinct time scales. Similarly, the gating mechanism (Yao et al., 2022) has been proposed as a means to modulate information storage and retrieval. Furthermore, researchers introduce a self-attention mechanism that allows temporal dependencies to be flexibly established (Yao et al., 2021; Qin et al., 2023). Nevertheless, these approaches either demonstrate limited efficacy in establishing long-term temporal dependencies (Bellec et al., 2018; Yin et al., 2021; Fang et al., 2021b; Yao et al., 2022) or they encounter challenges in terms of computational cost and hardware implementation (Yao et al., 2021; Qin et al., 2023).

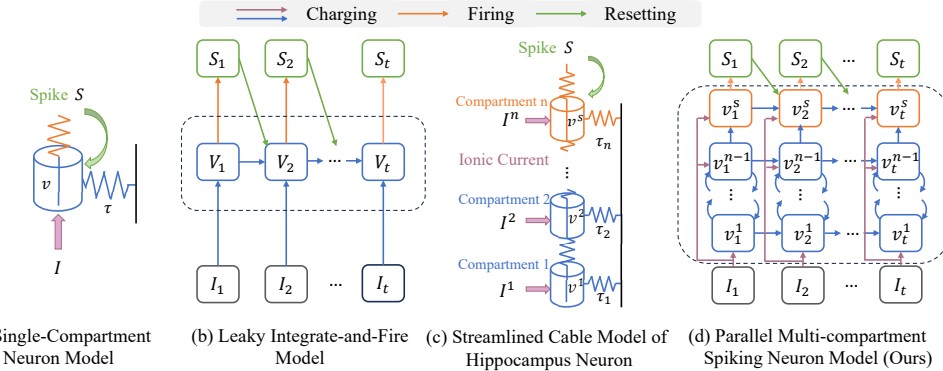

Figure 1: Comparison of structure and neuronal dynamics between simplified single-compartment neuron and bio-inspired multi-compartment neuron models. **(a, b)** The widely used Leaky Integrate-and-Fire model simplifies the biological neurons into a single compartment, resulting in deficiencies in multi-scale sequential modeling. **(c)** In comparison, the biological neuron involved in sensory processing and memory comprises multiple membrane regions with distinct characteristics, which can be faithfully represented as a multi-compartment cable model (Traub et al., 1991). **(d)** Our proposed Parallel Multi-compartment Neuron model adheres to the multi-compartment structure observed in (c), offering extended neuronal dynamics that are crucial for effective sequential modeling.

Multi-compartment neuron models have attracted significant attention due to their exceptional ability to capture the complex dynamics of biological neurons involved in spatiotemporal sensory signal processing and memory formation (Hines, 1984; Spruston, 2008; Markram, 2006). These models, with their rich temporal dynamics, can facilitate the representation and process of temporal information across distinct time scales (Burgess et al., 2002; Yu et al., 2021). Additionally, multi-compartment models are also compatible with emerging neuromorphic computing infrastructures (Yang et al., 2019). Despite their established significance in computational neuroscience, the application of these models in deep SNNs and pattern recognition tasks remains largely unexplored. Recent research has led to the development of two-compartment spiking neuron models, showing promise in sequential modeling (Shaban et al., 2021; Zhang et al., 2023). Nonetheless, it remains unexplored how to effectively scale these handcrafted two-compartment models to include additional neuronal compartments, while concurrently enhancing their sequential modeling capacity. Moreover, these models can only be updated sequentially in time, posing difficulties in training speed, especially when dealing with long sequences.

In this work, we propose a Parallel Multi-compartment Spiking Neuron (PMSN) model, a solution designed to unleash the potential of multi-compartment neurons in SNNs to solve sequential modeling. The proposed PMSN model, illustrated in Figure 1, exhibits diverse neuronal dynamics, which are essential for effective multi-scale sequential modeling. Furthermore, we introduce a parallel implementation strategy for the proposed model, which effectively decouples the non-linear dependencies among different time steps. This parallel implementation offers significant improvements in training speed, particularly on GPU-accelerated machine learning (ML) frameworks. The main contributions of this work are summarized as follows:

- We introduce a novel generalized multi-compartment spiking neuron model for SNNs. This model is specifically designed to encapsulate the intricate dynamics that exist among different neuronal compartments and demonstrates a remarkable capacity for multi-scale sequential modeling.

- We propose a parallel implementation for the proposed multi-compartment spiking neuron to enable efficient training on GPU-accelerated ML frameworks. Notably, our implementation takes into account the vital reset mechanism of spiking neurons, which has been disregarded in previous parallel spiking neuron models. The theoretical analysis on the gradient flow of the proposed model illuminates its efficacy in establishing temporal dependency across varying time scales.

- We conduct comprehensive experiments to validate the efficacy of the proposed PMSN model across diverse sequential modeling tasks related to perception. The results demonstrate superior accuracy, substantial acceleration in simulation speed, and favorable trade-off between computational cost and accuracy. Further visualization of multi-compartment dynamics validates its capacity to preserve and integrate multi-scale temporal information.

## 2 RELATED WORKS

**Memory-Enhanced Spiking Neuron Models.** Recent studies have introduced novel spiking neuron models that exhibit an enhanced capacity for sequential modeling. These models can be categorized into single- or multi-compartment models. For single-compartment models, recent studies have explored the incorporation of adaptively decaying variables to enhance the memory capacity of spiking neurons. For instance, Bellec et al. (2018) and Yin et al. (2021) propose the usage of adaptive firing thresholds, which function as long-term memory to enhance sequential modeling. Fang et al. (2021b) suggests the utilization of a learnable time constant for spiking neurons, resulting in a heterogeneous neuronal population capable of representing multi-scale temporal information. Additionally, Yao et al. (2022) introduces a gating mechanism into spiking neurons to explicitly control memory storage and retrieval processes. For multi-compartment models, Shaban et al. (2021) propose a two-compartment model where the thresholds undergo a double-exponential decay, facilitating the storage of both short-term and long-term information. Zhang et al. (2023) propose a two-compartment model that simulates the interactive dynamics between the soma and dendrites. Despite their enhanced capacity, their inherent restrictions to effective scaling and training speed still remain to be explored.

**Parallel Training Methods for SNNs.** Recently, researchers have become increasingly concerned about the slow training speed of SNNs. This issue arises due to the non-linear state-dependent nature of SNNs, which hinders parallel computation in the temporal dimension. Consequently, the full potential of GPU acceleration remains underutilized. To tackle this challenge, Fang et al. (2023) introduced a series of parallel spiking neural (PSN) models. These models transform the charge dynamics of the membrane potential into a learnable decay matrix and bypass the neuron's reset mechanism to enable parallel computation. However, these models require to access inputs beyond just the preceding time step, which is both biologically implausible and unsupported by current neuromorphic chips. While these models demonstrate superior performance in tasks involving short-term dependency, their single-compartment structure limits their capacity to model long-term dependency. Furthermore, the crucial neuronal reset mechanism has been ignored in this work.

**State Space Model.** The state space model (SSM), originally developed to formulate the dynamic of linear time-invariant (LTI) systems, has demonstrated exceptional performance in sequential modeling and presents a solution for parallel training of stateful recurrent models (Gu et al., 2022; Smith et al., 2023). While SSMs can be used to capture the membrane potential charging process of spiking neurons, the dynamics associated with non-linear spike generation and membrane potential reset, which are inherent to spiking neurons, necessitate special treatment. Notably, these aspects have been specifically addressed in the proposed PMSN model.

## 3 PRELIMINARIES

**Single-Compartment Spiking Neuron Models.** Inspired by the rich temporal dynamics of biological neurons, a number of single-compartment spiking neuron models have been proposed for large-scale brain simulation and neuromorphic computing (Gerstner et al., 2014). Among these models, the Leaky Integrate-and-Fire (LIF) model (Burkitt, 2006) has been the most frequently used one, which provides a well-balanced trade-off between biological plausibility and computational efficiency. The dynamics of its membrane potential $v$ can be formulated as follows:

$$\frac{dv(t)}{dt} = -\frac{1}{\tau_m}(v(t) - v_{reset}) + I(t), \qquad \text{if} \quad v(t) \geq \theta, \quad v(t) \to v_{reset}, \qquad (1)$$

The temporal dynamics of the above LIF model consist of three distinct phases: **charging, firing, and resetting**. During the charging phase, the information contained in the input current $I$ is integrated into membrane potential $v$ at a decay rate governed by $\tau_m$. Once $v$ exceeds the firing threshold $\theta$, an output spike will be generated and transmitted to subsequent neurons. Following the spike generation, the membrane potential will be reset to its initial value $v_{reset}$. In practice, the above continuous-time formulation is typically discretized using the Euler method as:

$$\begin{cases} V[t] = \alpha V[t-1] + I[t] - \theta S[t-1], \\ S[t] = H\left(V[t], \theta\right), \end{cases} \qquad (2)$$

where $H(\cdot)$ is the Heaviside function and $\alpha = e^{-\frac{dt}{\tau_m}}$ is the decaying rate. Despite its promising results in tasks involving limited temporal context, the LIF model encounters the following two

challenges when dealing with long sequences. Firstly, this single-compartment model struggles to maintain information over an extended time scale. Thus, it fails to establish long-term dependencies. Secondly, the required simulation time grows substantially with the sequence length, which is due to the non-linear state-dependent nature of the neuronal update. Specifically, the membrane potential update of $V[t]$ depends on the output spike $S[t-1]$ of the preceding time step, which is only available after $t-1$, as $S[t-1]$ has a non-linear dependency on $V[t-1]$. This time-coupled relationship prevents the membrane state from unfolding in time, leading to challenges in parallelization.

## 4 METHODS

In this section, we first present a generalized multi-compartment spiking neuron model for SNNs that abstracts the electrophysical properties of biological neurons. Based on that, we further develop the PMSN model which can effectively process multi-scale temporal information. In addition, to address the training speed concern of the multi-compartment model, we enhance this model to support parallel training on GPU-accelerated ML frameworks. Finally, we analyze how error gradients are effectively propagated in our proposed model to achieve temporal credit assignment.

### 4.1 A GENERALIZED MULTI-COMPARTMENT SPIKING NEURON MODEL

Biological neurons, with their sophisticated neural activities, excel in processing sequential sensory information across different time scales. For instance, neural oscillations have been shown to be crucial for memory formation in hippocampal pyramidal neurons (Jensen & Colgin, 2007). Similarly, cortical pyramidal neurons use oscillatory activities to synchronize and integrate sensory signals (Buzsáki & Wang, 2012). To accurately capture the electrophysical properties of these neurons, a variety of multi-compartment cable models has been extensively studied (Traub et al., 1991; Pinsky & Rinzel, 1994; Spruston, 2008). These models divide the membrane regions of single neurons into different neuronal compartments and allow current to flow through the neighboring ones. The interaction among these compartments facilitates the generation of varied and intricate neuronal activities. To enrich the temporal dynamics of deep SNN and explore its potential in sequential modeling, we abstract a generalized multi-compartment spiking neuron model from these numerical models. Notably, our model can be flexibly extended to incorporate any desired number of compartments, thereby accommodating the temporal complexity of different sequential modeling tasks. The neuronal dynamics, depicted in Figure 1(d), can be formulated as follows:

$$
\begin{cases}
\frac{dv^{(1)}(t)}{dt} = -\frac{1}{\tau_1}v^{(1)}(t) + \beta_{2,1}v^{(2)}(t) + \gamma_1 I(t), \\
\frac{dv^{(2)}(t)}{dt} = -\frac{1}{\tau_2}v^{(2)}(t) + \beta_{3,2}v^{(3)}(t) + \beta_{1,2}v^{(1)}(t) + \gamma_2 I(t), \quad \text{if} \quad v^{(n)}(t) \geq \theta, \; s(t) = 1, \\
\quad \cdots \qquad\qquad\qquad\qquad\qquad\qquad\qquad\qquad\qquad\qquad\qquad\qquad v^{(n)}(t) \to v^{(n)}(t) - \theta, \\
\frac{dv^{(n)}(t)}{dt} = -\frac{1}{\tau_n}v^{(n)}(t) + \beta_{n-1,n}v^{(n-1)}(t) + \gamma_n I(t),
\end{cases}
\tag{3}
$$

where $v^i$ represents the membrane potential of the compartment $i$, $\theta$ is the firing threshold, and $I(t)$ denotes the synaptic current transduced from the input spikes from the previous layer, mathematically expressed as $I^l(t) = \mathcal{W}^l S^{l-1}(t)$. Here, $\mathcal{W}$ is the synaptic strength between layer $l-1$ and $l$. Once the membrane potential of the final compartment $v^{(n)}$ exceeds the firing threshold $\theta$, an output spike will emit. Subsequently, the membrane potential will be reset following the reset-by-subtraction mechanism. The neuronal dynamic parameters, namely $\tau_i$, $\gamma_i$, and $\beta_{i,j}$, signify the membrane capacitance, current gain of the compartment $i$, and coupling strength between nearby compartments $i$ and $j$, respectively. These parameters are learned jointly with synaptic parameters $\mathcal{W}^l$ during training. It's important to note that, the compartments are only allowed to interact with their neighboring ones so as to reduce the overall model complexity. We could also rewrite these first-order differential equations into a $n$-order state space function as:

$$
\dot{\mathcal{V}}(t) = \begin{bmatrix}
-\frac{1}{\tau_1} & \beta_{2,1} & 0 & \cdots & 0 \\
\beta_{1,2} & -\frac{1}{\tau_2} & \beta_{3,2} & \cdots & 0 \\
\vdots & \vdots & \vdots & \ddots & \vdots \\
0 & 0 & \cdots & -\frac{1}{\tau_{n-1}} & \beta_{n,n-1} \\
0 & 0 & \cdots & \beta_{n-1,n} & -\frac{1}{\tau_n}
\end{bmatrix} \mathcal{V}(t) + \begin{bmatrix}
\gamma_1 \\
\gamma_2 \\
\vdots \\
\gamma_{n-1} \\
\gamma_n
\end{bmatrix} I(t) - \begin{bmatrix}
0 \\
0 \\
\vdots \\
0 \\
1
\end{bmatrix} \theta S(t),
\tag{4}
$$

where $\mathcal{V} = [v^{(1)}, v^{(2)}, , \cdots, v^{(n)}]^T$, and $S(t) = H(v^{(n)}(t), \theta)$.

## 4.2 PMSN: A Parallel Multi-compartment Spiking Neuron

It is worth noting that the model described above, when trained serially with BPTT, consumes significantly more simulation time than existing single-compartment spiking neuron models. Therefore, it becomes crucial to adapt this model to support parallel training, whose primary limitation lies in the utilization of a non-linear spike generation function, denoted as $H(\cdot)$. After incorporating the reset mechanism, this function introduces a non-linear temporal dependency between $\mathcal{V}(t + dt)$ and $\mathcal{V}(t)$. To surmount this limitation, we propose a **Parallel Multi-compartment Spiking Neuron (PMSN)** model based on Eq. 4, which allows such non-linear temporal dependency to be decoupled. We achieve this by setting $\beta_{n,n-1}$ to 0 and decoupling $n$ compartments into $n - 1$ hidden compartments $\mathcal{V}_h$ with linear recurrence, and one output compartment $v^{(n)} = v_s$ with nonlinear reset. The resulting neuronal dynamics of PMSN can be represented as follows:

$$
\dot{\mathcal{V}}_h(t) = \begin{bmatrix} -\frac{1}{\tau_1} & \beta_{2,1} & 0 & \cdots & 0 \\ \beta_{1,2} & -\frac{1}{\tau_2} & \beta_{3,2} & \cdots & 0 \\ \vdots & \vdots & \vdots & \ddots & \vdots \\ 0 & 0 & \cdots & \beta_{n-2,n-1} & -\frac{1}{\tau_{n-1}} \end{bmatrix} \mathcal{V}_h(t) + \begin{bmatrix} \gamma_1 \\ \gamma_2 \\ \vdots \\ \gamma_{n-1} \end{bmatrix} I(t), \tag{5}
$$

$$
\dot{v}_s(t) = \begin{bmatrix} 0 & 0 & \cdots & \beta_{n-1,n} \end{bmatrix} \mathcal{V}_h(t) - \frac{1}{\tau_n} v_s(t) + \gamma_n I(t) - \theta S(t), \quad S(t) = H(v_s(t), \theta), \tag{6}
$$

where the hidden compartments are designated to form a $(n-1)$-order cable model, while the output compartment is charged by the last hidden compartment but resets independently. In the following, we first explain how the linear recurrent states $\mathcal{V}_h$ can be computed in parallel, and the non-linear temporal dependency issue associated with $v_s$ will be addressed subsequently.

**Parallel Implementation for Hidden Compartments.** The multi-compartment spiking neuron model presented above forms a non-linear continuous system, we first apply the zero-order hold (ZOH) method (DeCarlo, 1989) to temporally discretize the continuous-time formulations as stated in Eqs. 5 and 6. Specifically, we utilize a full-rank state transition matrix $\mathcal{T} \in \mathbb{R}^{n-1 \times n-1}$ to represent the first matrix in Eq. 5, which could be diagonalized using eigenvalue decomposition $\mathcal{T} = P\Lambda P^{-1}$, where $\Lambda$ is the diagonal eigenvalue matrix, and $P \in \mathbb{C}^{n-1 \times n-1}$ denotes eigenvector matrix. Consequently, we can obtain the following discrete-time formulation:

$$
V_h[t] = \bar{\mathcal{T}} V_h[t-1] + \Phi_c I[t], \tag{7}
$$

$$
I_h[t] = \Phi_s V_h[t] + \gamma_n I[t], \tag{8}
$$

$$
v_s[t] = \alpha v_s[t-1] + I_h[t] - \theta S[t-1], \quad S[t] = H(v_s[t], \theta), \tag{9}
$$

where $V_h = P^{-1}\mathcal{V}_h$, $\bar{\mathcal{T}} = exp(\Lambda dt)$, $\Phi_c = \Lambda^{-1}(exp(\Lambda dt) - I)\phi_c$, $\phi_c = P^{-1}[\gamma_1, .., \gamma_{n-1}]^T$. $I_h[t]$ signifies the total input current to the output compartment, $\Phi_s = [0, .., \beta_{n-1,n}]P$, $\alpha = exp(-\frac{dt}{\tau_n})$. The values of $\Lambda dt$, $\phi_c$, $\gamma_n$, and $\Phi_s$, which are derived from the neuronal dynamic parameters, are all learnable. The model in Eq. 7 is an LTI system, which could be considered to own a linear recurrence. Alternatively, we could unfold it as:

$$
V_h[t] = \sum_{i=0}^{t} \bar{\mathcal{T}}^{t-i} \Phi_c I[i]. \tag{10}
$$

We further define a kernel $\mathcal{K} = [\Phi_s \bar{\mathcal{T}}^{t-1} \Phi_c, ..., \Phi_s \bar{\mathcal{T}}^0 \Phi_c]$. In this way, the Eq. 8 can be simplified to perform matrix multiplication with input set $\boldsymbol{I_t} = \{I[0], I[1], ...I[t]\}$ as:

$$
I_h[t] = \mathcal{K} \cdot \boldsymbol{I_t} + \gamma_n I[t]. \tag{11}
$$

Notably, the above computation can be parallelized over time by applying the convolution operation to the first term as:

$$
I_h[0:t] = \mathcal{K} * \boldsymbol{I_t} + \gamma_n \boldsymbol{I_t} = \mathcal{F}^{-1}(\mathcal{F}(\mathcal{K}) \cdot \mathcal{F}(\boldsymbol{I_t})) + \gamma_n \boldsymbol{I_t}, \tag{12}
$$

where $\mathcal{F}$, $\mathcal{F}^{-1}$ signifies forward and inverse Fourier Transform respectively. In this way, we could efficiently compute the membrane potential for the first $n-1$ hidden compartments $V_h$ and the input for the output compartment $I_h$ across all timesteps in parallel. However, how to obtain the output spike train in parallel remains to be addressed.

**Parallel Implementation for Output Compartment with Reset.** The output compartment $v^s$ is responsible for spike generation and reset. However, this compartment could not apply the parallel implementation due to the existence of the non-linear reset. To be specific, as given in Eq. 9, we must apply the non-linear function $H(\cdot)$ on $v_s[t-1]$ to obtain spike output $S[t-1]$ before computing the membrane potentials $v_s[t]$. This non-linearity prevents the unfolding of iterative functions into linear mathematical expressions over time, thus restricting the temporal parallelization of the model. To resolve this issue, we try to circumvent computing the non-linear dependency between timesteps. Specifically, we approach this problem by focusing on obtaining the cumulative inputs and resets. To this end, we further refine the step-by-step dynamics of the output compartment in Eq. 9 as:

$$v_s[t] = \alpha v_s[t-1] + I_h[t] - v_r[t-1], \quad S[t] = H(v_s[t], \theta), \tag{13}$$

$$v_r[t] = \theta S[t] \cdot \lfloor v_s[t] \rfloor_\theta, \tag{14}$$

where $v_r[t]$ denotes the membrane potential that has been subtracted during reset. $\lfloor \cdot \rfloor_\theta$ signifies the floor division by $\theta$. Compared with the original reset-by-subtraction mechanism described in Eqs. 6 and 9, we force $v_s$ to reset to a level below threshold $\theta$, which bears closer resemblance with the repolarization process of biological neurons. The parallel computation of the output compartment is achieved by setting the decay constant $\alpha = 1$ and clamping the input $I_h$ to non-negative before aggregating it over time. The detailed derivations are provided in the Appendix. Thus, we could unfold the iterative dynamics in Eq.13 and eliminate the dependency between $v_s[t]$ and $S[t-1]$ as:

$$\sum_{i=0}^{t-1} v_r[i] = \theta \lfloor \sum_{i=0}^{t-1} I_h[i] \rfloor_\theta, \quad v_s[t] = \sum_{i=0}^{t} I_h[i] - \theta \lfloor \sum_{i=0}^{t-1} I_h[i] \rfloor_\theta. \tag{15}$$

The cumulative sum of input current set $\boldsymbol{I_{h,t}} = \{\sum_{i=0}^{0:t} I_h[i]\}$, can be efficiently computed using the parallel prefix sum (Scan) algorithm (Harris et al., 2007). Therefore, we could derive the output spike train set $\boldsymbol{S_t} = \{S[0], S[1], ...S[t]\}$ based on the obtained membrane potential as:

$$\boldsymbol{S_t} = H\left(\boldsymbol{I_{h,t}} - \theta \lfloor \boldsymbol{I_{h,t-1}} \rfloor_\theta\right). \tag{16}$$

Notably, this reset mechanism can also be seamlessly generalized to the non-leaky single-compartment models, thereby enabling parallelizable training. The generalization is straightforward, and only requires replacing the input $I_h$ with the input current of the single-compartment model.

## 4.3 Effective Temporal Gradient Propagation

Here, we provide a theoretical analysis to explain how gradients can effectively propagate to earlier times in our PMSN model to facilitate multi-scale sequential modeling. The detailed derivation process can refer to the Appendix. To overcome the discontinuity that happened during spike generation and reset, we employ surrogate gradients (Deng et al., 2022; Bengio et al., 2013) and result in a unique gradient flow for the PMSN parameter update as:

$$\Delta \mathcal{W}^l \propto \frac{\partial \mathcal{L}}{\partial \mathcal{W}^l} = \sum_{t=1}^{T} \frac{\partial \mathcal{L}}{\partial I^l[t]} S^{l-1}[t], \quad \Delta b^l \propto \frac{\partial \mathcal{L}}{\partial b^l} = \sum_{t=1}^{T} \frac{\partial \mathcal{L}}{\partial I^l[t]},$$

$$\frac{\partial \mathcal{L}}{\partial I^l[t]} = \sum_{i=t}^{T} \frac{\partial \mathcal{L}}{\partial S^l[i]} \frac{\partial S^l[i]}{\partial v_s^l[i]} \frac{\partial v_s^l[i]}{\partial I_h^l[i]} \frac{\partial I_h^l[i]}{\partial I^l[t]} = \underbrace{\frac{\partial \mathcal{L}}{\partial S^l[i]} g'[t]\gamma_n}_{\text{Spatial}} + \underbrace{\sum_{i=t}^{T} \frac{\partial \mathcal{L}}{\partial S^l[i]} g'[i]\Phi_s \bar{\mathcal{T}}^{i-t}\Phi_c}_{\text{Temporal}}, \tag{17}$$

where $g'[t] = \frac{\partial S^l[i]}{\partial v_s^l[i]}$ is the surrogate gradient function, $\mathcal{L}$ is the loss function. $\mathcal{W}^l$ and $b^l$ refer to weight and bias terms of layer $l$, respectively. The first term of the final gradient represents the gradient spatial propagation, while the second term signifies the gradient propagation in the temporal domain. Note that the proposed PMSN model possesses multiple neuronal compartments with varying decay rates, denoted as $\bar{\mathcal{T}} = \text{diag}(\lambda_1, ...\lambda_{n-1})$. This can effectively establish temporal dependencies across varying time scales with a proper set of values for $\bar{\mathcal{T}}$. Furthermore, the gradient update remains unaffected by the neuronal reset. In contrast, the vanilla spiking neurons, when utilizing the BPTT method for temporal credit assignment, encounter challenges in learning long-term dependency. This is attributed to the vanishing gradient problem caused by recursive membrane potential decay and reset.

Table 1: Comparison of classification accuracy of different models on long-term sequential tasks.

| Dataset | Timesteps | Approach | Parallel Training | Architecture | Parameters | Accuracy |
|---|---|---|---|---|---|---|
| S-MNIST & PS-MNIST | 784 | LIF (Zhang et al., 2023) | N | Feedforward | 85.1k | 72.06% / 10.00% |
| | | PLIF (Zhang et al., 2023) | N | Feedforward | 85.1k | 87.92% / N.A. |
| | | GLIF (Zhang et al., 2023) | N | Feedforward | 87.5k | 95.27% / N.A. |
| | | DEXAT (Shaban et al., 2021) | N | Recurrent | N.A. | 96.40% / N.A. |
| | | ALIF (Yin et al., 2021) | N | Recurrent | 156.3k | 98.70% / 94.30% |
| | | TC-LIF (Zhang et al., 2023) | N | Recurrent | 155.1k | 99.20% / 95.36% |
| | | SPSN (Fang et al., 2023)* | Y | Feedforward | 52.4k | 97.20% / 82.84% |
| | | masked PSN (Fang et al., 2023)* | Y | Feedforward | 153.7k | 97.76% / 97.53% |
| | | PSN (Fang et al., 2023)* | Y | Feedforward | 2.51M | 97.90% / 97.76% |
| | | **Ours** | **Y** | **Feedforward** | **66.3k** | **99.40% / 97.16%** |
| | | | | | **156.4k** | **99.53% / 97.78%** |
| SHD | 250 | Adaptive axonal delay (Sun et al., 2023) | N | Feedforward | 109.1k | 92.45% |
| | | TA-SNN (Yao et al., 2021) | N | Feedforward | 121.7k | 91.08% |
| | | ALIF (Yin et al., 2021) | N | Recurrent | 141.3k | 84.40% |
| | | TC-LIF(Zhang et al., 2023) | N | Recurrent | 141.8k | 88.91% |
| | | LIF (Zenke & Vogels, 2021) | N | Recurrent | 249k | 84.00% |
| | | ASGL (Wang et al., 2023) | N | Feedforward | 230.4k | 87.90% |
| | | RadLIF (Bittar & Garner, 2022) | N | Feedforward | 3.9M | 94.62% |
| | | SPSN (Fang et al., 2023)* | Y | Feedforward | 107.1k | 82.51% |
| | | masked PSN (Fang et al., 2023)* | Y | Feedforward | 122.5k | 86.00% |
| | | PSN (Fang et al., 2023)* | Y | Feedforward | 232.5k | 89.75% |
| | | **Ours** | **Y** | **Feedforward** | **120.3k** | **94.25%** |
| | | | | | **199.3k** | **95.10%** |

\* Our reproduced results based on publicly available codebases     N.A.  These results are not publicly available

# 5 EXPERIMENTS

In this section, we evaluate the proposed PMSN model, focusing on its multi-scale sequential modeling capacity, simulation acceleration, and computational cost efficiency. Unless otherwise stated, all tested PMSNs have $n = 5$ compartments to ensure a comparable computational cost to the state-of-the-art (SOTA) parallel spiking neuron models (i.e., 32-receptive-field sliding parallel spiking neuron (SPSN) (Fang et al., 2023)). Detailed configurations are provided in the Appendix, and we will make our code publicly available after the review process.

## 5.1 ESTABLISHING TEMPORAL DEPENDENCIES ACROSS DISTINCT TIME SCALES

In Table 1, we first compare our PMSN model against other SOTA models on sequential modeling tasks involving **long-term dependencies**. Our experiments are conducted on three widely used benchmarks, including Sequential MNIST (S-MNIST) and Permuted Sequential MNIST (PS-MNNIST) datasets with 784 time steps (Le et al., 2015), and Spiking Heidelberg Digits (SHD) spoken digit classification dataset with 250 time steps. Notably, our PMSN models achieve the highest accuracies across all these tasks, with fewer or comparable amounts of parameters, demonstrating a superior capacity to establish long-term dependency. Furthermore, the single-compartment models, due to their limited memory capacity, generally perform worse than the multiple-compartment ones, including DEXAT (Shaban et al., 2021) and TC-LIF (Zhang et al., 2023). Additionally, our model performs substantially better than the recently introduced parallel spiking neuron models PSN, masked PSN, and SPSN (Fang et al., 2023) that are constrained with a single compartment.

We further evaluate our model on establishing **spatiotemporal and extended long-term temporal dependencies** through more challenging tasks: Sequential CIFAR10 and CIFAR100. For Sequential CIFAR10, we explore two configurations: column-by-column scanning as per Fang et al. (2023) ($T = 32$), which evaluates the model's capacity in integrating both spatial and temporal information, and pixel-by-pixel scanning ($T = 1024$), which poses a greater challenge for learning long-term dependency. For Sequential CIFAR100, we use the column configuration. To ensure a fair comparison, we employ the same network architecture for each individual task. As shown in Table 2, our PMSN model surpasses the SOTA models by at least 2% accuracy, showcasing its superiority in multi-scale sequential modeling. As provided in Figure 2, the learning curves of the PMSN model exhibit faster and more stable training convergence for the reasons that have been explained in Section 4.3.

Furthermore, we would like to stress the importance of the **neuronal reset mechanism** that has been neglected in (Fang et al., 2023). As presented in Table 2, the accuracy consistently improves for both PMSN and LIF models after incorporating the reset mechanism. By preventing the membrane potential from becoming excessively high, the reset mechanism yields a smoother distribution of membrane potentials across time, thereby facilitating a more stable information flow and training.

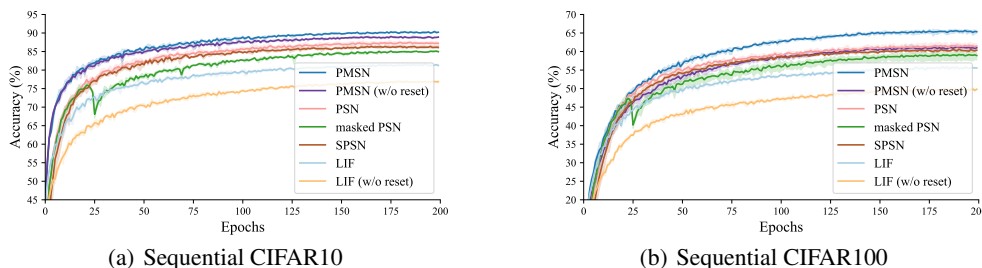

(a) Sequential CIFAR10          (b) Sequential CIFAR100

Figure 2: Learning curves of different neurons on column-level tasks from three independent runs.

Table 2: Comparing distinct neurons to learn spatiotemporal and extended long-term dependencies.

| Tasks | Timesteps | PMSN | PMSN (w/o reset) | PSN | masked PSN | SPSN | LIF | LIF (w/o reset) |
|---|---|---|---|---|---|---|---|---|
| Sequential CIFAR10 | | **90.97%** | 89.27% | 88.45% | 85.81% | 86.70% | 81.50% | 79.50% |
| Sequential CIFAR100 | 32 | **66.08%** | 60.82% | 62.21% | 60.69% | 62.11% | 55.45% | 53.33% |
| Parameters | | 0.54M | 0.54M | 0.52M | 0.52M | 0.51M | 0.51M | 0.51M |
| Sequential CIFAR10 | 1024 | **82.14%** | 79.63% | 55.24% | 57.83% | 70.23% | 45.07% | 43.30% |
| Parameters | | 206.9k | 206.9k | 6.47M | 376.7k | 177.1k | 176.9k | 176.9k |

## 5.2 VISUALIZATION OF MULTI-COMPARTMENT DYNAMICS

Despite the superior performance of the PMSN model, it remains unclear how it integrates multi-scale temporal information through the intertwined dynamics of neuronal compartments. To address this concern, we first perform a single-neuron analysis and visualize the dynamics of each compartment in Figure 3. Specifically, we input an impulse at $T = 0$ and record the trajectory of the membrane potential across compartments. The **Left** figure shows the interaction of hidden compartments results in damped oscillations of membrane potentials, each with a distinct oscillation frequency and decaying speed characterized by its dynamic coefficient $\lambda_i = e^{\alpha + \beta i}$. These oscillations, whose duration is proportional to $e^{-\alpha}$, allow the input signals to be preserved across different time scales. Moreover, the variance in oscillation frequency and decay among different compartments suggests the PMSN model can effectively integrate information across different frequency domains and time spans, thereby enabling multi-scale sequential modeling. Moving beyond single-neuron analysis, we also plot the distribution of oscillation frequency and damping coefficients across neuron populations within one layer. The results provided in the **Middle** and **Right** figures reveal diverse compartmental temporal characteristics across the neuron population, which enhances the network capacity in multi-scale sequential modeling. See the comprehensive analysis in the Appendix.

## 5.3 SIMULATION ACCELERATION

To quantify the simulation acceleration of the proposed parallel implementation, we record its actual inference and training times using the GPU-enabled Pytorch library and compare them against a serial implementation that achieves identical neuronal dynamics. Figure 4 illustrates the acceleration ratios across various model sizes and sequence lengths. Our parallel implementation significantly advances the serial one, achieving speed-up ratios ranging from 5.4 to 217 and 6.2 to 302 for for-

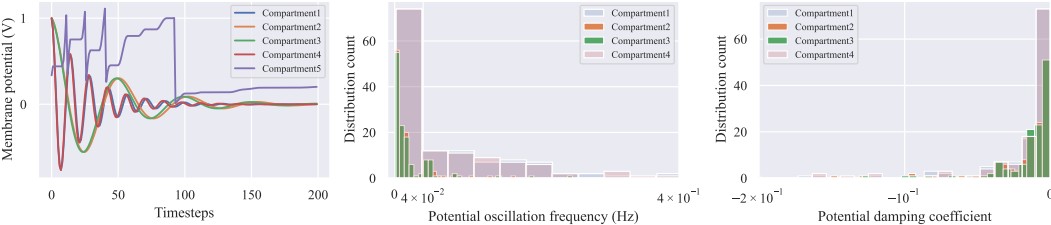

Figure 3: **Left:** The impulse response of different compartments within the same neuron. The transient response time (i.e., the time interval from the input start to the cease of oscillation) of the hidden membrane potential indicates how fast the information decays in each compartment. **Middle:** The distribution of oscillation frequencies $\beta/2\pi$, and **Right:** damping coefficients $\alpha$ for different hidden compartments of neurons in the selected layer. These two figures suggest the PMSN model can effectively integrate temporal information across different frequency domains and time scales.

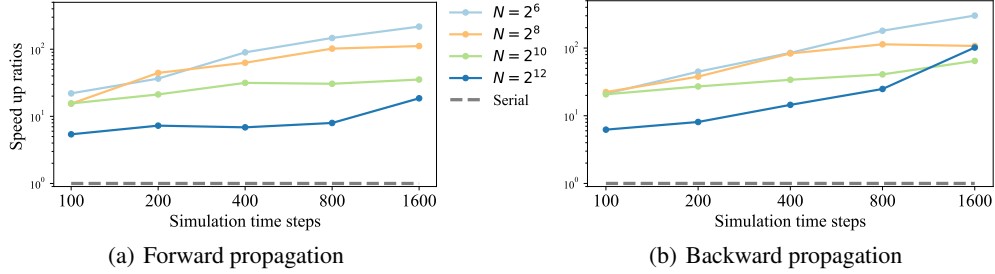

(a) Forward propagation  (b) Backward propagation

Figure 4: The simulation speed-up ratios $t_{serial}/t_{PMSN}$ achieved by the proposed parallel implementation, compared to a serially updated identical model, where $N$ denotes the total neuron count.

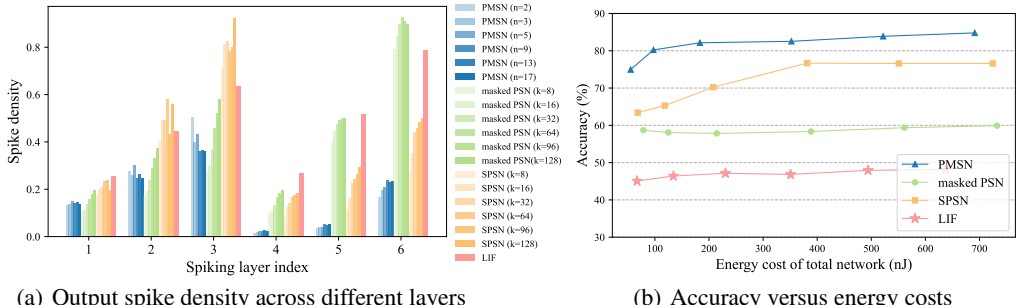

(a) Output spike density across different layers  (b) Accuracy versus energy costs

Figure 5: Comparison among LIF and different parallel spiking neuron models. Calculation processes of the energy cost are provided in the Appendix.

ward (inference) and backward (training) propagation, respectively. Moreover, we notice a positive correlation between the speed-up ratio and sequence length, favorable for long sequences.

## 5.4 COMPUTATIONAL COST

The computational cost during inference remains a key concern for the practical use of the proposed PMSN model. To address this concern, we conduct a comparative study to assess the computational cost of the PMSN model, the PSN families (Fang et al., 2023), and the LIF model. The experiments are conducted on the pixel-by-pixel Sequential CIFAR10 task, employing the same network structure for all models. It is important to note that the total computational cost is affected by both neuronal and synaptic updates. In the case of our PMSN model, scaling up the total number of compartments $n$ leads to a nearly linear increase in the cost of neuronal updates. Similarly, enlarging the temporal receptive field, denoted as "k", in the k-order masked PSN and SPSN models has the same effect. In contrast, the cost of synaptic updates primarily depends on the sparsity of output spikes. Notably, as illustrated in Figure 5(a), our proposed PMSN model achieves the lowest spike density, which can potentially be attributed to the inhibitory effect resulting from the coupling compartments. We further plot the accuracy-energy curves for different spiking neuron models in Figure 5(b). Each point on these curves corresponds to a specific model from Figure 5(a). The PSN model, which has a considerably high cost of $5541 \ nJ$, is omitted from this figure. Our PMSN model consistently outperforms other models in terms of accuracy when provided with the same amount of energy. Notably, the accuracy of the PMSN model exhibits rapid improvement as the number of compartments increases from 2 to 5, while the improvement plateaus beyond that point. Therefore, it offers users the flexibility to strike a favorable balance between computational cost and accuracy.

## 6 CONCLUSION

This paper presents a novel multi-compartment neuron model with enhanced capacity in multi-scale sequential modeling. Additionally, a parallel implementation is provided for the proposed multi-compartment neuron model, which enables parallel training on GPU-accelerated ML frameworks. Our experimental results demonstrate its superior performance in establishing temporal dependencies across distinct time scales, acceleration, and accuracy-energy trade-offs. It, therefore, opens up a bountiful of opportunities for solving sequential modeling tasks with neuromorphic solutions.

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
