# OpenReview forum: "A Parallel Multi-compartment Spiking Neuron For Multi-scale Sequential Modeling"
_ICLR.cc/2024/Conference — Submitted to ICLR 2024_

### Official Review · Reviewer_Chwq · 2023-10-28

**Soundness:** 2 fair
**Presentation:** 3 good
**Contribution:** 2 fair
**Rating:** 6
**Confidence:** 2

**Summary:**

The authors propose a new architecture for sequence modelling, based on multicompartment neuron dynamics. They then develop a method for parallel training of these systems and apply them to several benchmark tasks.

**Strengths:**

The experimental results seem very strong, significantly outperforming previous methods. In addition, the figures are clear and help the reader with understanding the content. Lastly, parallelizing the training of such spiking multi-compartment models in the temporal dimensions is novel (to my knowledge) and can be potentially impactful.

**Weaknesses:**

**Soundness**

One of the main contributions of the paper, the parallel implementation of the algorithm, seems to hinge on the fact that they set beta_{n, n-1}. What is the effect of this on the neuronal dynamics? Can this model still be considered biophysically realistic?

**Clarity**

Many parts of the paper are presented in an overly convoluted way. I believe that this paper would largely benefit from moving math that is not essential to understanding the context of the paper to the appendix, for example equations 17, 18, 19.

In addition, it would be valuable if the authors attributed a biophysical meaning to their learnable parameters.

Finally, I fail to understand where equation 14 comes from (although I am not exactly from the field, maybe it is clear to other reviewers).

**Novelty**

I believe that the claims on novelty for the multicompartment model are a bit over-stated. At the very least should the authors cite some of the (decades of) work on multicompartment modelling and its numerical implementations (starting from Hines 1984). The voltage update equations (apart from the non-linear reset) are identical to those papers, and I think this should be clarified.

**Questions:**

See questions in the "weaknesses" section.

Page 6: I do not understand this sentence: `we force vs to reset to a level below the firing threshold, which bears closer resemblance with biological observations.` Why does this have closer resemblance to biological observations? Also closer than what other mechanism?

Lastly, the authors claim (even in the abstract) that their work is motivated by a cable model of cable model of hippocampus pyramidal neurons. Given that the model they eventually use is a 5-compartment cable without any particular dynamics, this choice of model seems extremely specific. What exactly is the reason to claim that the model is in any way "hippocampal" or even "pyramidal"?

---

> ### Author Response · Authors · 2023-11-15
> **Response to Reviewer Chwq (1/2)**
>
> > **Soundness 1:** One of the main contributions of the paper, the parallel implementation of the algorithm, seems to hinge on the fact that they set beta_{n, n-1}. What is the effect of this on the neuronal dynamics? Can this model still be considered biophysically realistic?
>
> **Response:** Thanks for your insightful comment. First of all, we would like to clarify that this work lies in the field of neuromorphic computing, which aims to design brain-inspired neural architecture that can solve real-world pattern recognition tasks with improved energy efficiency and effectiveness. This differs from the studies in computational neuroscience that aim to simulate and study the dynamics of biologically realistic spiking neuronal systems. Secondly, as you have rightly pointed out, we set $\beta_{n, n-1}$ to 0 in our model as it is a necessary setting to enable model parallelization across the temporal dimension. From the neuronal dynamics perspective, this operation eliminates the feedback effect from the $n^{th}$ compartment to the ${n-1}^{th}$ compartments. However,  its impact should not be viewed in isolation. During the training process, all other compartments adapt and evolve,  effectively compensating for the effects of this abstraction. As clearly demonstrated in Figure 3 and Appendix A.6, the learned dynamics within each compartment exhibit rich and varied multi-scale temporal patterns. These dynamics confirm the effectiveness of our proposed multi-compartment neuron model in capturing complex temporal dependencies across various time scales as their biological counterparts.
>
> &nbsp;
>
> > **Clarity 1:** Many parts of the paper are presented in an overly convoluted way. I believe that this paper would largely benefit from moving math that is not essential to understanding the context of the paper to the appendix, for example equations 17, 18, 19.
>
> **Response:** Thanks for your helpful suggestion. We will move Equations 18 and 19 to Appendix, and thoroughly improve the clarity of the paper in our revised manuscript.
>
> &nbsp;
>
> > **Clarity 2:** In addition, it would be valuable if the authors attributed a biophysical meaning to their learnable parameters.
>
> **Response:** We really appreciate your suggestions. The learnable parameters in our network can be categorized into two parts. The first part is the weight matrix $\mathcal{W}$ between network layers, which can be interpreted as synaptic strength between neurons. The second part consists of the parameters within the proposed neuron models. More specifically, as defined in Equation 3, $\beta$ can be considered as the coupling strength between different compartments, while $\tau$ is the membrane capacitance. These neuronal parameters are trained jointly with weight parameters in our SNNs. To improve the clarity, we will supplement the biophysical meaning of these parameters in our revised manuscripts to provide a more intuitive understanding of our formulation.
>
> &nbsp;
>
> > **Clarity 3 & Question 1:** Finally, I fail to understand where equation 14 comes from (although I am not exactly from the field, maybe it is clear to other reviewers). Page 6: I do not understand this sentence: we force vs to reset to a level below the firing threshold, which bears closer resemblance with biological observations. Why does this have closer resemblance to biological observations? Also closer than what other mechanism?
>
> **Response:** Thank you for raising concerns on this point. In Equations 13 and 14, we have described the neuronal dynamics of the last compartment that takes charge of spike generation. Inspired by the repolarization process of biological neurons after each spike generation, the reset-by-subtraction scheme [1] has been commonly used in SNNs to reset the membrane potential after neuron firing. In particular, the membrane potential ($v_s$) is reset by subtracting the value of the threshold ($\theta$) from it. In contrast, our model modifies this mechanism to enable parallelization for the output compartment. This is achieved by subtracting the value of $ \theta \lfloor v_s/\theta \rfloor$ from $v_s$ as given in Equation 14. This operation ensures the membrane potential has been reset to a level that is below the firing threshold as well as close to the resting potential ($0$).
> Let me provide an example to illustrate the difference between these two reset schemes. For instance, assume $v_s[t]=10.5$, $\theta=1$, and the resting potential = 0. Following the reset-by-subtraction scheme, $v_s[t+1]$ will be 9.5 after resetting, which is still much higher than the threshold. In contrast, following our reset scheme, $v_s[t+1]$ will be 0.5, which is closer to the resting potential. Therefore, our model more closely resembles the repolarization process observed in biological neurons, which reset the membrane potential to a value close to its resting potential.

---

> > ### Author Response · Authors · 2023-11-15
> > **Response to Reviewer Chwq (2/2)**
> >
> > > **Novelty 1:** I believe that the claims on novelty for the multicompartment model are a bit over-stated. At the very least should the authors cite some of the (decades of) work on multicompartment modelling and its numerical implementations (starting from Hines 1984). The voltage update equations (apart from the non-linear reset) are identical to those papers, and I think this should be clarified.
> >
> > **Response:** We agree with you that the multi-compartment model and its numerical implementations have been widely studied in the neuroscience field. As mentioned in the Introduction (Page 2, Paragraph 1) and Section 4.1 of our manuscript, we have referenced some of the studies on the multi-compartment model, acknowledging their contributions and the inspiration we have borrowed from. Following your suggestions, we will further improve our manuscript by adding refs to other canonical works (e.g., Hines 1984) along this direction in the revised version.
> >
> > However, in the field of neuromorphic computing that we are focusing on, the study of multi-compartment neuron models within deep SNNs remains **extremely limited**. This point is also mentioned by reviewer 9iDX: *``This model is not new, however, it is largely restricted to biological spiking networks instead of SNNs’’.* To bridge this research gap, we propose a multi-compartment spiking neuron model to improve the sequential modeling capacity of existing SNNs.
> > Specifically, our novelty lies in two aspects:
> >
> > 1. As an early attempt to incorporate the concept of multi-compartment models into deep SNNs to solve real-world tasks, to our best knowledge, we are the **first** model that incorporates more than two compartments. Our proposed multi-compartment model **significantly enhanced the capacity of SNNs in dealing with multi-scale sequential signals** when compared with other SNN neuron models. We have included typical state-of-the-art spiking neuron models that have been applied to SNNs for comparison.
> >
> > 2. We provide a **GPU-accelerated solution to re-engineer this model to enhance the simulation speed**. This provides at most hundreds of times speed up over the serial version and makes it practical to solve more challenging real-world sequential modeling tasks. This is also the **first** multi-compartment model in SNNs that supports parallel training on GPUs.
> >
> > &nbsp;
> >
> > > **Question 2:** Lastly, the authors claim (even in the abstract) that their work is motivated by a cable model of cable model of hippocampus pyramidal neurons. Given that the model they eventually use is a 5-compartment cable without any particular dynamics, this choice of model seems extremely specific. What exactly is the reason to claim that the model is in any way "hippocampal" or even "pyramidal"?
> >
> > **Response:** Thank you for bringing this up. As we have mentioned in the Introduction, due to the oversimplified spiking neuron models used in neuromorphic computing research, current SNNs suffer from the poor memory capacity that is critical for solving real-world sequential modeling tasks. To bridge this gap, we seek inspiration from the hippocampus pyramidal neurons that are **known for their crucial role in memory formation**. Specifically, we borrow ideas from the branch of studies that use the multi-compartment model to describe the dynamics of these hippocampus pyramidal neurons. However, these models suffer from high computational complexity that prohibits them from being implemented in deep SNNs and solving challenging real-world pattern recognition tasks. For instance, [2] used 19 compartments to capture the neuronal dynamics of CA3 hippocampal pyramidal neurons. Subsequently, [3] simplified the model into 2 compartments. In this work, we take inspiration from this line of research and propose a general multi-compartment neuron model that can strike a balance between model complexity and the richness of neuron dynamics required for sequential modeling tasks. We further introduce GPU-accelerated implementation for this model to achieve superior training and inference efficiency on deep learning infrastructure.
> >
> > We sincerely hope this has addressed your concern.
> >
> > &nbsp;
> >
> > **Reference**
> >
> > [1]Rueckauer, B., Lungu, I. A., Hu, Y., Pfeiffer, M., & Liu, S. C. (2017). Conversion of continuous-valued deep networks to efficient event-driven networks for image classification. Frontiers in neuroscience, 11, 682.
> >
> > [2]Traub, R. D., Wong, R. K., Miles, R., & Michelson, H. (1991). A model of a CA3 hippocampal pyramidal neuron incorporating voltage-clamp data on intrinsic conductances. Journal of neurophysiology, 66(2), 635-650.
> >
> > [3]Pinsky, P. F., & Rinzel, J. (1994). Intrinsic and network rhythmogenesis in a reduced Traub model for CA3 neurons. Journal of computational neuroscience, 1, 39-60.

---

> > > ### Comment · Reviewer_Chwq · 2023-11-18
> > > **Thank you for the response!**
> > >
> > > Thank you for the thorough responses. It clarifies all concerns apart from the last one: why is the model in any way "hippocampus" or "pyramidal"?
> > >
> > > Your response gives two reasons, namely that (1) they have a crucial role for memory formation and (2) you borrow ideas from multicompartment models of hippocampus pyramidal studies.
> > >
> > > I am not convinced by this:
> > > (1) They do indeed play a role in memory formation, but why is this so crucial for your work? E.g. cortical pyramidal cells are extremely important for sensory processing. Why is memory formation closer to the tasks in your paper than sensory processing?
> > >
> > > (2) which ideas do you borrow that only relate to "hippocampus pyramidal studies"? Multicompartment models have been built for an extremely diverse set of neuron types and it unclear to me why relate less to your work than multicompartment hippocampus pyramidal studies.

---

> > > > ### Author Response · Authors · 2023-11-19
> > > > **Thanks for your further suggestions, here is our response. (1/2)**
> > > >
> > > > Dear reviewer Chwq,
> > > >
> > > > Thanks for your valuable feedback regarding our emphasis on hippocampus pyramidal models in this work. Your suggestion has prompted a thoughtful reevaluation of how we present our model's inspirations. Upon further reflection, we understand that the multicompartment dynamics we've abstracted, although inspired by hippocampus pyramidal neurons, are not unique to this specific neuron type. We really appreciate you have pointed out this problem and it helps us to improve the clarity and rigor of our manuscript.
> > > >
> > > > **Memory formation or Sensory Processing:** In the context of work, we primarily focus on addressing the limitations in memory capacity of existing SNN models. Specifically, when input signals are integrated into the membrane potential of simplified spiking neurons (i.e., LIF model), their memory traces undergo rapid decay and resetting. Consequently, it causes difficulties in establishing long-term dependencies between input cues and distant reward/punishment signals over extended time spans. We fully agree with you on the point that this capacity is also essential for sensory processing, and indeed we have used several sensory processing tasks to evaluate the superior sequential modeling capacity of our proposed multi-compartment neuron model.
> > > >
> > > > **Manuscript Refinement:** In response to your comments, we will revise our manuscript to highlight the general significance of multicompartment models. We will clarify that our inspiration is drawn from a general concept of multicompartment cable models, which are extensively used to simulate the complex temporal dynamics of biological neurons engaged in spatiotemporal sensory signal processing and memory formation. These models, with their rich temporal dynamics, can facilitate the representation and process of temporal information across distinct time scales. While these models are well-explored in computational neuroscience, their use in deep SNNs and pattern recognition tasks remains largely unexplored. We believe that incorporating interactive dynamics among compartments, as we propose, can significantly enhance SNNs' ability to process temporal signals with multi-scale temporal dynamics. It, therefore, will open up many exciting opportunities for neuromorphic computing. Specifically, we will make revisions in the following sections:
> > > >
> > > > 1. In the **Abstract**, we will revise to ``We propose a novel Parallel Multi-compartment Spiking Neuron (PMSN), inspired by the multi-compartment model that simulates the temporal dynamics of biological neurons involved in sensory processing and memory.``
> > > >
> > > > 2. In **Introduction Para. 3**, we will clarify that `` Multicompartment neuron models have attracted significant attention due to their exceptional ability to capture the complex dynamics of biological neurons involved in both spatiotemporal sensory signal processing and memory formation [1, 2, 3]. These models, with their rich temporal dynamics, can facilitate the representation and process of temporal information across distinct time scales [4, 5]. Additionally, these models are also compatible with emerging neuromorphic computing infrastructures [6]. While these models are well-explored in computational neuroscience, their use in deep SNNs and pattern recognition tasks remains largely unexplored. ``
> > > >
> > > > 3. In **Section 4.1**, we will modify the statement of inspiration to ``Biological neurons, with their sophisticated neural activities, excel in processing sequential sensory information across different time scales. For instance, neural oscillations have been shown to be crucial for memory formation in hippocampal pyramidal neurons [7]. Similarly, cortical pyramidal neurons use oscillatory activities to synchronize and integrate sensory signals [8]. To accurately capture the electrophysical properties of these neurons, a variety of multi-compartment models has been extensively studied [1, 9, 10].``
> > > >
> > > > We hope that these revisions will thoroughly address your concerns. Again, we are grateful for your constructive suggestions, which have significantly contributed to the enhancement of our work.

---

> > > > > ### Author Response · Authors · 2023-11-19
> > > > > **Thanks for your further suggestions, here is our response. (2/2)**
> > > > >
> > > > > **Reference:**
> > > > >
> > > > > &nbsp;
> > > > >
> > > > > [1] Spruston, N. (2008). Pyramidal neurons: dendritic structure and synaptic integration. Nature Reviews Neuroscience, 9(3), 206-221.
> > > > >
> > > > > [2] Hines, M. (1984). Efficient computation of branched nerve equations. International journal of bio-medical computing, 15(1), 69-76.
> > > > >
> > > > > [3] Markram, H. (2006). The blue brain project. Nature Reviews Neuroscience, 7(2), 153-160.
> > > > >
> > > > > [4] Burgess, N., Maguire, E. A., & O'Keefe, J. (2002). The human hippocampus and spatial and episodic memory. Neuron, 35(4), 625-641.
> > > > >
> > > > > [5] Yu, D., Wang, G., Li, T., Ding, Q., & Jia, Y. (2023). Filtering properties of Hodgkin–Huxley neuron on different time-scale signals. Communications in Nonlinear Science and Numerical Simulation, 117, 106894.
> > > > >
> > > > > [6] Yang, S., Deng, B., Wang, J., Li, H., Lu, M., Che, Y., ... & Loparo, K. A. (2019). Scalable digital neuromorphic architecture for large-scale biophysically meaningful neural network with multi-compartment neurons. IEEE transactions on neural networks and learning systems, 31(1), 148-162.
> > > > >
> > > > > [7] Jensen, O., & Colgin, L. L. (2007). Cross-frequency coupling between neuronal oscillations. Trends in cognitive sciences, 11(7), 267-269.
> > > > >
> > > > > [8] Buzsáki, G., & Wang, X. J. (2012). Mechanisms of gamma oscillations. Annual review of neuroscience, 35, 203-225.
> > > > >
> > > > > [9] Traub, R. D., Wong, R. K., Miles, R., & Michelson, H. (1991). A model of a CA3 hippocampal pyramidal neuron incorporating voltage-clamp data on intrinsic conductances. Journal of neurophysiology, 66(2), 635-650.
> > > > >
> > > > > [10] Pinsky, P. F., & Rinzel, J. (1994). Intrinsic and network rhythmogenesis in a reduced Traub model for CA3 neurons. Journal of computational neuroscience, 1, 39-60.

---

> > > > > ### Comment · Reviewer_Chwq · 2023-11-19
> > > > > **Thank you!**
> > > > >
> > > > > Thank you for the response and proposed modifications! This clarifies my concerns and I have increased my score.

---

### Official Review · Reviewer_kRbB · 2023-10-30

**Soundness:** 2 fair
**Presentation:** 3 good
**Contribution:** 2 fair
**Rating:** 3
**Confidence:** 5

**Summary:**

This paper proposes the parallel multi-compartment spiking neuron (PMSN). The PMSN has n compartments, and only the last compartment can fire and reset. Thus, the neuronal dynamics of the first (n - 1) compartments can be paralleled easily. For the last compartment, the soft reset and the floor function are combined, which parallelizes the neuronal dynamics with reset. The performance and energy estimation are validated on temporal datasets.

**Strengths:**

1. Compared with the previous work PSN (Fang et al., 2023) which removes the neuronal reset, the PMSN can be parallelized with the neuronal reset. The ablation experiments show that the removal of neuronal reset decreases task performance.

2. The experiment results on sequential CIFAR are high, showing the advantage of the PMSN.

**Weaknesses:**

In section 5.3, only the speeds of serial and parallel implementation of the PMSN are compared.

The accuracy of the ImageNet dataset, which is an important benchmark for deep SNNs, is not reported.

**Questions:**

Can the authors provide a speed comparison between the PSN and the PMSN?

What are the theoretical FLOPs and memory consumption of the PMSN? I suggest that the authors provide a comparison between these two neurons. It would be better if a comparison between the memory consumption during training of an SNN with two neurons is provided.



--post rebuttal

Thanks for your response. It is clear that the PMSN is more similar to the sliding PSN rather than the PSN. I agree with reviewer 9iDX that the current model only applies to the last compartment. The accuracy of the model is lower than the compared PSN method on the ImageNet dataset. Also, it is a pity that the current implementation of the PMSN does not make full use of GPUs.
Thus, I decide to keep my ranking.

---

> ### Author Response · Authors · 2023-11-15
> **Response to Reviewer kRbB (1/2)**
>
> > **Weakness 1 & Question 1, 2:** In section 5.3, only the speeds of serial and parallel implementation of the PMSN are compared. Can the authors provide a speed comparison between the PSN and the PMSN? What are the theoretical FLOPs and memory consumption of the PMSN? I suggest that the authors provide a comparison between these two neurons. It would be better if a comparison between the memory consumption during training of an SNN with two neurons is provided.
>
> **Response:** Thanks a lot for raising concerns about the insufficient comparison of speed, computational cost, and memory consumption between PSN and PMSN. We have already conducted both empirical and theoretical studies on the computational cost of PMSN and PSN models. Our theoretical analysis, detailed in Appendix A.5, breaks down the theoretical computational cost into two primary components: the number of Multiply-Accumulate (MAC) operations and the number of cheap Accumulate (AC) operations. In our analysis, the FLOPs correspond to the number of MAC operations. Consider a scenario with  $m$ neurons across a time window of length $t$. The theoretical FLOPs for PSN and PMSN (5-compartment) are calculated as $mt^2$, and $32mt$, respectively. This indicates that as the time window lengthens, the PSN model requires increasingly more FLOPs compared to the PMSN model.
>
> As per your suggestion, we have further conducted a comparative analysis focusing on memory consumption and training/inference speed using the following three benchmarks. In all experiments, identical training configurations and network architecture are used. The results are provided as follows:
> | Dataset  | S-MNIST (T=784) | Sequential CIFAR10 (T=32) | Sequential CIFAR10 (T=1024) |
> | ----------------- | --------------- | ------------------------- | --------------------------- |
> | **Training time (seconds/epoch)** ||||
> | PMSN | 35.3235 | 73.2060  | 61.7562|
> | PSN | 21.1776 | 49.0240 | 34.0634 |
> | **Testing time (seconds/epoch)**  ||||
> | PMSN | 2.6903 | 6.4175 | 4.7766 |
> | PSN | 1.4773 | 4.1030  | 2.4646  |
> | **Memory consumption (GB)**  ||||
> | PMSN | 1.4385 | 3.1369  | 9.4000|
> | PSN | 0.6130 | 1.7618  | 9.0833 |
> | **Accuracy (%)**  ||||
> | PMSN |  99.40 | 90.97 | 82.14 |
> | PSN |  97.90 | 88.45 | 55.24 |
>
> As expected, the PMSN model requires more time and memory than the PSN model across all datasets. This increased demand is primarily attributed to a larger number of neuronal compartments being used (i.e., five in PMSN v.s. one in PSN). However, the increase in actual time and memory consumption is relatively modest, generally less than twice that of the PSN model in all experiments. This highlights the effectiveness of our proposed parallelization method. Furthermore, we believe this additional computational cost is worthwhile when considering the significantly enhanced sequential modeling capacity as demonstrated on the Sequential Cifar10 and Cifar100 datasets.
>
> To further strengthen our manuscript, we will include these new results and discussions in our revised manuscript to provide a more comprehensive comparison between the two models.

---

> > ### Author Response · Authors · 2023-11-15
> > **Response to Reviewer kRbB (2/2)**
> >
> > > **Weakness 2:** The accuracy of the ImageNet dataset, which is an important benchmark for deep SNNs, is not reported.
> >
> > **Response:** Thanks for your suggestion. We fully agree that ImageNet is an important benchmark for deep SNN research. However, we would like to point out that ImageNet primarily serves as a benchmark for assessing the efficacy of **spatial credit assignment or spatial dependency modeling** of SNNs. This distinction is crucial, as our paper focuses on the efficacy of **temporal credit assignment and temporal dependency modeling**. The static nature of the ImageNet dataset limits its utility in this context, as it only contains spatial information. Therefore, it fails to fully leverage the inherent temporal dynamics of spiking neurons.
> >
> > Although it's possible to convert ImageNet into a sequential format, as has been done for the CIFAR10 and CIFAR100 datasets, the required computational resource would be significantly boosted compared to the training on the static image. Specifically, we estimate the training time on a GPU server equipped with 8 Nvidia 3090 cards would span several months for a single model. Additionally, it's important to note the absence of an established SNN benchmark for this specific task. Given these considerations, including the limited relevance of the ImageNet dataset to our research focus and the considerable computational demands, we have decided to exclude it from comparison.
> >
> > Following the same research direction to enhance the temporal dependency modeling of SNNs, several impactful studies have developed SNNs with diverse structures and adaptive neuron models [1-4]. In this work, we adhere to their experimental setting and rigorously evaluate our model on a comprehensive set of sequential modeling benchmarks that allow for direct comparison with existing models.
> >
> > We hope this response has properly addressed your concern.
> >
> > &nbsp;
> >
> > **Reference**
> >
> > [1] Bellec, G., Salaj, D., Subramoney, A., Legenstein, R., & Maass, W. (2018). Long short-term memory and learning-to-learn in networks of spiking neurons. Advances in neural information processing systems, 31.
> >
> > [2] Bellec, G., Scherr, F., Subramoney, A., Hajek, E., Salaj, D., Legenstein, R., & Maass, W. (2020). A solution to the learning dilemma for recurrent networks of spiking neurons. Nature communications, 11(1), 3625.
> >
> > [3] Yin, B., Corradi, F., & Bohté, S. M. (2021). Accurate and efficient time-domain classification with adaptive spiking recurrent neural networks. Nature Machine Intelligence, 3(10), 905-913.
> >
> > [4] Yao, M., Gao, H., Zhao, G., Wang, D., Lin, Y., Yang, Z., & Li, G. (2021). Temporal-wise attention spiking neural networks for event streams classification. In Proceedings of the IEEE/CVF International Conference on Computer Vision (pp. 10221-10230).

---

> > > ### Comment · Reviewer_kRbB · 2023-11-16
> > > **Thanks for your response. I have more questions.**
> > >
> > > 1. The PMSN has lower FLOPs than the PSN, but its training speed is slower. Can you explain the reason?
> > >
> > > 2. The PMSN consumes more memory during training. What is the number of parameters of the PMSN?
> > >
> > > 3. I do not fully agree with your response about ImageNet. Although ImageNet is a static dataset, the training of SNNs on this dataset usually uses T > 1. If we use the Real-Time Recurrent Learning (RTRL) method to train SNNs on ImageNet, the accuracy also drops. We can infer that temporal credit assignment and temporal dependency modeling are also required during the training on the ImageNet dataset.
> > >
> > > Moreover, the under-fitting problem is the main issue of training SNNs on the ImageNet dataset, while the over-fitting problem is the main issue on other datasets, such as the neuromorphic CIFAR10-DVS and SHD datasets. This is also the reason why the ImageNet dataset has become a well-acknowledged benchmark in deep learning.
> > >
> > > With 8x 3090 GPUs, dozens of epochs can be trained during the rebuttal period. Although the SNNs will not be trained well, the accuracy of the early epochs can also be reported and compared (between the PMSN and the PSN).

---

> > > > ### Author Response · Authors · 2023-11-21
> > > > **Thanks for your further comment, here is our response (1/2).**
> > > >
> > > > Dear reviewer,
> > > >
> > > > Thanks for your prompt and insightful comments. We would like to address your concern with the following response:
> > > >
> > > > &nbsp;
> > > >
> > > > > **Question 1:** The PMSN has lower FLOPs than the PSN, but its training speed is slower. Can you explain the reason?
> > > >
> > > > **Response:** Thank you for raising this question. While our theoretical analysis suggests that the PMSN model, with lower inference FLOPs, might be potentially faster in training speeds than the PSN model, its **parallel training implementation presents some efficiency challenges** that affect its training speed.
> > > >
> > > > The core of the PMSN training speed issue lies in its innovative approach to enable parallel training on GPU frameworks. Our model, adhering to the recurrent nature of SNNs, ensures that neurons access input only from their current or previous time steps during training and inference to enable applicability in real-time processing. Consequently, we employ forward and inverse Fourier transformation (as detailed in Eq. 12) to achieve temporal convolution during parallel training. By contrast, the PSN model allows neurons access to input across all time steps, including future ones, a scenario that diverges from real-time processing constraints. While this approach sacrifices the neuromorphic applicability that our PMSN model upholds, it enables the PSN model to leverage simple tensor multiplication during parallel training.
> > > >
> > > > Unfortunately, the current implementations of the *torch.fft* and *torch.ifft* functions in the PyTorch framework, **do not parallelize as efficiently as the tensor multiplication operations** utilized in the PSN model. This difference in the computational efficiency of underlying operations is a significant factor contributing to the slower computational speed of the PMSN model in practice.
> > > >
> > > > Notably, there is a promising avenue for **improvement**. We could use *`associative_scan’* function provided by the Triton library to replace these Fourier transformations, which is expected to provide a more efficient training speed. While the exact extent of this improvement remains to be explored, it is a promising direction for our future work.
> > > >
> > > > &nbsp;
> > > >
> > > > > **Question 2:** The PMSN consumes more memory during training. What is the number of parameters of the PMSN?
> > > >
> > > > **Response:** We really appreciate your comments. As detailed in our experimental results, the PMSN model has a **parameter count** of 66.3k, 0.54M, and 206.9k for S-MNIST, Sequential CIFAR10 (T=32), and Sequential CIFAR10 (T=1024) tasks, respectively. In comparison, the PSN model owns a parameter count of 2.51M, 0.52M, and 6.47M for these datasets. This notable gap in parameter counts can be attributed to the fact that, the temporal parameter count of PSN scales with the square of the time window ($T^2$). Conversely, the parameter count of PMSN remains consistent regardless of the time window length. This fundamental architectural difference underlines the more **efficient parameterization** of the PMSN model, particularly in scenarios involving longer time windows.
> > > >
> > > > While our analysis intuitively suggests that the PMSN model, with its fewer parameters, might occupy less memory during training, it's crucial to note that the higher memory consumption observed in the PMSN model is not due to its parameter count. Instead, it stems from the **sophisticated structure of its recurrent computational graph**, as depicted in Figure 1(d). Our PMSN model maintains enriched memory states within each neuron to restore the previous memory traces effectively. However, this enhancement introduces a more intricate computational flow within these memory states. During the forward and backward propagation processes, these enriched memory states generate a multitude of intermediate results, which need to be temporarily stored in memory, thereby leading to the observed increase in memory occupation. In contrast, the PSN model only applies a fully connection layer in neuronal state computation across all timesteps. This design spares memory during training by avoiding the storage of intermediate results. However, it is important to note that this simplicity comes at a cost of significant computational cost during serial inference and is inaccessible to real-time processing.
> > > >
> > > > Ultimately, it’s important to recognize that our PMSN model, strikes a balance between computational requirements (ratio less than 2) and its augmented sequential modeling capabilities. Moreover, as PMSN only access to the input from current and previous timesteps, it is promising to spare more memory cost used for storing previous inputs than PSN when implemented to neuromorphic chips for serial inference.

---

> > > > > ### Author Response · Authors · 2023-11-21
> > > > > **Thanks for your further comment, here is our response (2/2).**
> > > > >
> > > > > >**Question 3: ImageNet experiments.** We can infer that temporal credit assignment and temporal dependency modeling are also required during the training on the ImageNet dataset. Moreover, the under-fitting problem is the main issue of training SNNs on the ImageNet dataset, while the over-fitting problem is the main issue on other datasets, such as the neuromorphic CIFAR10-DVS and SHD datasets. This is also the reason why the ImageNet dataset has become a well-acknowledged benchmark in deep learning. With 8x 3090 GPUs, dozens of epochs can be trained during the rebuttal period. Although the SNNs will not be trained well, the accuracy of the early epochs can also be reported and compared (between the PMSN and the PSN).
> > > > >
> > > > > **Response:** Thank you for your constructive feedback. Inspired by your insights, we have enthusiastically applied our PMSN model to the ImageNet dataset to assess its capabilities in the realm of static image classification. Despite the challenging time constraints of the rebuttal period, we have made **significant efforts** to implement PMSN model on this dataset and managed to train the PSN/PMSN models for 55 epochs (320 in total), respectively. We believe these initial results, although preliminary, shed light on the potential and areas for improvement of our model in static image classification.
> > > > >
> > > > > To ensure a comprehensive and fair comparison, we have employed the SEW ResNet-18 network architecture, SGD optimizer, CE loss (excluding TET), and a time window length of T=4 for both sets of experiments. These early-stage results, including detailed training/testing curves, are now **available in the supplementary materials**, revealing that the PMSN model converges slightly slower than the PSN model. Specifically, while PSN achieves a classification accuracy of 64.26% at Epoch 55, PMSN records 63.32%. Interestingly, the accuracy curves show that PMSN initially converges faster than PSN, but over time, PSN exhibits better convergence.
> > > > >
> > > > > We are proactively investigating the underlying factors contributing to this phenomenon. Potential causes might include an unsuitable learning rate, inappropriate initial settings of the PMSN model that are better suited for long-term sequential tasks (e.g., slow decay rate, strong coupling coefficient), or issues related to the overly rich temporal dynamics impacting the flatness of loss landscape. We believe that these issues might be addressed with adjustments such as employing TET loss to smooth the loss landscape. While we regret not being able to present more conclusive results within the rebuttal timeframe, it's noteworthy that our early findings already **surpass the performance of the SEW ResNet-18 model with LIF neurons** (63.18%), underscoring the PMSN model's effective use of complex temporal dynamics.
> > > > >
> > > > > We are grateful for your suggestions, which have undoubtedly helped enhance the quality of our research. These preliminary results suggest that the PMSN model is not only effective in sequential modeling across various timescales but also holds promise in static image classification tasks within limited time windows. This broadens the potential application of our work beyond just sequential tasks. We are committed to resolving the current issues and will complete a more comprehensive set of experiments, including those with SEW ResNet-18 and SEW ResNet-34, for our camera-ready manuscript.
> > > > >
> > > > > We hope this hasty update addresses your concerns and demonstrates our dedication to enhancing our work. Thank you again for your insightful suggestions.

---

> > > > ### Author Response · Authors · 2023-11-23
> > > > **A Gentle Reminder for Final Hours for Discussion**
> > > >
> > > > Dear reviewer kRbB,
> > > >
> > > > &nbsp;
> > > >
> > > > Thank you for your time and effort in reviewing our paper. As we near the end of the discussion period, please let us know if you have any remaining questions or concerns. We have diligently addressed your previous comments, including results from new experiments.
> > > >
> > > > Feel free to reach out with any final feedback in the next few hours. Your insights are greatly valued. Thank you again for your thorough review and suggestions.
> > > >
> > > > &nbsp;
> > > >
> > > > Sincerely,
> > > >
> > > > Submission5407 Authors

---

### Official Review · Reviewer_9iDX · 2023-10-31

**Soundness:** 3 good
**Presentation:** 3 good
**Contribution:** 2 fair
**Rating:** 6
**Confidence:** 3

**Summary:**

The authors develop a new multi-compartment spiking neural model,
which is can be efficiently computed on GPUs, because 1) feed-forward
architecture is assumed, so that generated spikes at one time do not
effect the membrane potential in future and 2) the spiking activity is
decoupled from the membrane potential of all but the last compartment.
The authors show how the equations can be efficiently integrated and
compute on GPUs (as membrane potential evolution is just a linear
filter of the inputs). They compare the results on benchmarks
requiring long temporal integration of information, where typical SNNs
(in particular feed-forward SNNs based on simple
leaky-integrate-and-fire neurons) struggle, and show that the richer
compartmental dynamics indeed can capture long term information.

**Strengths:**

* Overall, it is a solid paper that suggests a new neuron model based on
compartments for SNNs trained with SGD, and shows in detail how to efficiently implement the
simulation in forward, backward, and gradient update. The idea of
dendritic computing is not new, however, it is largely restricted to
biological spiking networks instead of SNNs.

* Since the added dynamics is at the neuron-level (instead of synapse
level), and its efficient implementation, the additional computational
cost for simulating (or deploying on neuromorphic hardware) is
manageable.

**Weaknesses:**

* The
implementation seems to require feed-forward structure as well as
positive inputs, which seems quite restrictive (in particular the
latter).

* The neuron model does not seem to
improve the benchmark results dramatically over other approaches in
all tasks (ie recurrent RNNs are similar for S-MNIST) and it thus
remains open how useful the model will be.

**Questions:**

* I don't follow why one can assume that $I(t)$ is always positive for
the ``parallel implementation of the output compartment with
reset''. In typical SNNs, the input current is given by $I(t)=WS(t-1)$
(without a synapse model) and the weights are positive or negative
real numbers, so in general, the inputs *can* be negative. Isn't that
assumed here? However, if negative inputs are allowed, then the floor
of the accumulated input with the threshold $\theta$ (in Eq 15) will
not result in the number of spikes over the time period. Maybe I am
missing something here. If only positive currents are allowed, it
should be discussed as it seems to be a major restriction. If so it
would be interesting to know what the impact of this step would be (on
the simulation speed) if not done in parallel to allow for negative
currents.

* It is not clearly stated whether the flux parameters and decay rates
of and between the compartments are learned with SGD?

* In the equations (eg Eq. 3) the neural indices are written as
super-script ($v^{i}$) which is easily confused with power. Better to
use subscripts or write $v^{(i)}$ to avoid confusion.

---

> ### Author Response · Authors · 2023-11-15
> **Response to Reviewer 9iDX (1/2)**
>
> > **Weakness 1 & Question 1:** The implementation seems to require feed-forward structure as well as positive inputs, which seems quite restrictive (in particular the latter). I don't follow why one can assume that $I(t)$ is always positive for the ``parallel implementation of the output compartment with reset''.
>
> **Response:** We greatly appreciate your time in reviewing our manuscript and providing constructive feedback. As you have rightly pointed out, we did assume that $I_h[t]$ to be non-negative in Equation 15. This is a compromise to enable the parallelization of the reset mechanism, and we provide a detailed proof of Equation 15 based on this assumption in Appendix A.4.2.
> However, it's worth mentioning that this assumption only applies to the last compartment. It does not require the input to our PMSN model, namely $I(t)$, to be always positive. In fact, the negative input current to other compartments (i.e., 1 to n-1) in Equation 5  is allowed. As specified below the Equation 3, its value is formulated as $I^l(t)=\mathcal{W}^l S^{l-1}(t)$. Therefore, we **don’t need to impose any constraint on the input signal $I(t)$ of the PMSN model**, but only apply a clamp function on the intermediate states (output of Equation 12) before it is fed into Equation 15. We will highlight this point in our revised manuscript to avoid confusion.
>
> In the following, we would like to discuss the **impact of removing this assumption**. Firstly, according to our analysis, if negative inputs are allowed for the output compartment, the state update of the last compartment can only be done in a serial manner, while other compartments can still be computed in parallel. Following your suggestion, we have conducted an ablation study to quantify its impact on the simulation speed. The time elapse (seconds/epoch) for PMSN and last-compartment-serial PMSN are 62 vs 158 on sequential CIFAR-10 (T=32), and 73 vs 5182 on sequential CIFAR-10 (T=1024) tasks. These exploded simulation time underscore the necessity of the proposed parallel solution. Interestingly, we also observed that the last-compartment-serial PMSN model, which allows negative input for the last compartment, performs worse than the PMSN model. It exhibits a 3.98% and 4.5% accuracy drop on sequential CIFAR-10 and sequential CIFAR-100 (T=32) tasks, respectively. We suspect that negative input can result in negative membrane potentials that adversely affect the gradient flow. This is because when the membrane potential is far from the firing threshold, the gradient g’[t] computed from the surrogate function will become zero. This issue hinders the effective gradient propagation, therefore, leads to poorer classification accuracies than our proposed one.
>
> &nbsp;
>
> > **Weakness 2:** The neuron model does not seem to improve the benchmark results dramatically over other approaches in all tasks (ie recurrent RNNs are similar for S-MNIST) and it thus remains open how useful the model will be.
>
> **Response:**  We appreciate your insightful comment. While it is true that the performance enhancement of our model over recurrent SNNs may not seem dramatic in the context of the S-MNIST task, it’s worth noting that this task is relatively easy and models can perform well with limited memory capacity. The existing methods have already reached a very high accuracy and the margin for improvement is limited. Specifically, the state-of-the-art (SOTA) model TC-LIF with an RSNN architecture achieves an error rate of just 0.8%. Nevertheless, our model managed to reduce this error rate to 0.47% further.
>
> We understand that the S-MNIST dataset alone may not fully demonstrate the potential benefits of our proposed model. Thus, to further illustrate the enhanced sequential modeling capabilities of our proposed model, we have extended our comparisons to include more challenging sequential tasks. In these scenarios, our model consistently outperforms other SOTA models by significant margins. For instance, in the SHD speech command recognition task, our model achieves an accuracy improvement of 1.8% and 5.3% over the SOTA Adaptive Axonal Delay (feedforward) and TC-LIF (recurrent) model respectively, while maintaining a comparable number of parameters. Moreover, in the Sequential CIFAR10 (T=32, 1024) and Sequential CIFAR100 (T=32) tasks, our model surpasses the accuracy of other SOTA models by 2.52%, 11.91%, and 3.87%, respectively. It is worth noting that, the accuracy improvements increase along with the increasing complexity of the task.
>
> We believe these results underscore the potential of our model in handling more complex sequential modeling tasks, and we look forward to exploring its capabilities further in future research.

---

> > ### Author Response · Authors · 2023-11-15
> > **Response to Reviewer 9iDX (2/2)**
> >
> > > **Question 2:** It is not clearly stated whether the flux parameters and decay rates of and between the compartments are learned with SGD?
> >
> > **Response:** Thank you for pointing this out. In this work, the flux parameters and decay rates are learned jointly with weight parameters using AdamW Optimizer. We have mentioned this point below Equation 9, and we will further emphasize their biophysical meanings in the revised version of our manuscript. Furthermore, we will provide a detailed summary of our training configurations in our Appendix and make our code publicly available after the revision process to ensure reproducibility and facilitate understanding of our work.
> >
> > &nbsp;
> >
> > > **Question 3:** In the equations (eg Eq. 3) the neural indices are written as super-script ($v^i$) which is easily confused with power. Better to use subscripts or write $v^{(i)}$ to avoid confusion.
> >
> > **Response:** We sincerely appreciate your suggestion. We will revise these indices in our manuscripts to improve clarity.

---

> ### Comment · Reviewer_9iDX · 2023-11-19
>
> I thank the authors for the detailed response and clarification. Given that the assumptions of a positive current only applies to the last compartment and not to the others and has limited  (or even detrimental) effect on accuracy, I will revise my score to 6.

---

### Official Review · Reviewer_QBv1 · 2023-11-01

**Soundness:** 3 good
**Presentation:** 3 good
**Contribution:** 2 fair
**Rating:** 5
**Confidence:** 3

**Summary:**

The authors define a multi-compartment spiking neuron model that is able to process signals of "high temporal complexity".  Importantly, this model accounts for the spike reset mechanism.  The Parallel Multi-Compartment Spiking Neuron is inspired by hippocampal pyramidal neurons and admits a formulation that facilitates parallel training on GPUs.  The authors demonstrate the model's advantages in terms of performance and complexity.

**Strengths:**

The model and the approximations/trade-offs made are well-presented and clear.  Capturing complex temporal sequences is also a useful property.  A GPU accelerated model is of course an advantage in the modern day with the ubiquity of such hardware.

**Weaknesses:**

The model is very specific, without any particular justification.  When one uses spiking networks, it is usually (in my experience) because one wants to model a spiking neuron system, but then choices of compartments are no longer arbitrary.  This is really a question of motivation for the paper that seems missing to me.  What problem are the authors solving?  Why does one need a spiking neuron model? Or is this particular spiking neuron model related to one in common use in circumstances unfamiliar to me?  If it is already related to a model in common use, then having a fast implementation is of some importance.

A related issue is that the models used for comparisons seem very limited (if one is only concerned with whether the model is a spiking model).  Many groups have been training spiking neural networks for a long time (Zenke & Ganguli 2018 is but one example).  If the multi-compartment model itself is not of particular importance, but I needed to train a spiking model, I could just as well use a network model from one of these groups.  Does the PMCS have advantages over those models?

**Questions:**

The main question I have (from above) is why this particular model?

---

> ### Author Response · Authors · 2023-11-15
> **Response to Reviewer QBv1**
>
> **Response:**  We appreciate your comments and would like to address your concerns with our following responses:
>
> &nbsp;
>
> > **Weakness 1:**  The model is very specific, without any particular justification. When one uses spiking networks, it is usually (in my experience) because one wants to model a spiking neuron system, but then choices of compartments are no longer arbitrary.
>
> **Background –** Different from the studies in computational neuroscience that aim to simulate and study the dynamics of biologically realistic spiking neuronal systems, this work lies in the field of neuromorphic computing. In particular, we aim to design brain-inspired neural architecture that can solve real-world pattern recognition tasks with improved energy efficiency and effectiveness. To this end, simplified phenomenological spiking neuron models are typically used (e.g., Leaky Integrate-and-Fire (LIF) model), which allow SNNs to be implemented in silico with extremely low energy cost. In this context, the choice of compartments in our model is also driven by these goals. We trade off its computational efficiency and efficacy in sequential modeling tasks, making the model well-suited for processing complex temporal signals with low-energy requirements.
>
> &nbsp;
>
> > **Weakness 2:** This is really a question of motivation for the paper that seems missing to me. What problem are the authors solving? Why does one need a spiking neuron model? Or is this particular spiking neuron model related to one in common use in circumstances unfamiliar to me?
>
> **Research Gap –** While SNNs have achieved substantial performance improvements for tasks with limited temporal dynamics, such as image classification, their performance in handling tasks with long sequence lengths and rich temporal dynamics (e.g., audio signal processing) remains limited. This is attributed to the limited memory capacity and neuronal dynamics offered by the commonly used LIF model, which is also the one used in **(Zenke & Ganguli 2018)**. This oversimplified model, while computationally efficient, exhibits limited capacity in representing the rich neuronal dynamics of biological neurons. To bridge this gap, we design a memory-enhanced multi-compartment spiking neuron model in this work. Our model specifically addresses these limitations by leveraging the interplay among neuron compartments.
>
> &nbsp;
>
> > **Weakness 3:** If it is already related to a model in common use, then having a fast implementation is of some importance.
>
> **Novelty -** Our model draws inspiration from the multi-compartment model used to describe biological neurons. As mentioned by Reviewer 9iDX *``This model is not new, however, it is largely restricted to biological spiking networks instead of SNNs’’.* In the context of SNNs, our proposed model is the **first** to incorporate more than two compartments, demonstrating superior performance in sequential modeling. Furthermore, we introduce a GPU-accelerated solution to enhance the simulation speed of the proposed model. This is also the **first** multi-compartment model in SNNs that supports parallel training on GPUs.
>
> &nbsp;
>
> > **Weakness 4:** A related issue is that the models used for comparisons seem very limited (if one is only concerned with whether the model is a spiking model). Many groups have been training spiking neural networks for a long time (Zenke & Ganguli 2018 is but one example).
>
> **Outstanding Sequential Modelling Capacity -** In our experiment section, we aim to provide a holistic view of the existing SNN research by including all the state-of-the-art memory-enhanced spiking neuron models that have been studied earlier, including the LIF model that had been used in **(Zenke & Ganguli 2018)**. Our analysis suggests that the proposed multi-compartment model can not only be trained in parallel, but also excel in modeling multi-scale temporal dependencies, thereby bridging the performance gap between biological neuron and spiking neuron models in terms of the sequential modeling capacity.
>
> &nbsp;
>
> > **Weakness 5:** If the multi-compartment model itself is not of particular importance, but I needed to train a spiking model, I could just as well use a network model from one of these groups. Does the PMCS have advantages over those models?
>
> **Practicality –** As discussed above, if you're seeking to train an SNN to process temporal signals with long sequence length and rich temporal dynamics (e.g., in scenarios like speech processing), our model presents an excellent choice due to its enhanced sequential modeling capacity and rapid training speed.

---

> > ### Author Response · Authors · 2023-11-23
> > **A Gentle Reminder for Final Hours for Discussion**
> >
> > Dear reviewer QBv1,
> >
> >  &nbsp;
> >
> > As the discussion period nears its end, we thank you for your insightful feedback on our paper. We've carefully considered your comments and provided a detailed response above. Our work addresses key challenges in SNN performance for processing rich temporal dynamics, by introducing a novel, GPU-accelerated parallel multi-compartment model, achieving superior classification accuracy and speed acceleration in typical sequential modeling tasks. If there are any remaining questions or points for clarification, we are more than willing to address them in these final hours. Your expertise has been crucial to refining our work.
> >
> > Thank you again for your thorough review.
> >
> >  &nbsp;
> >
> > Sincerely,
> >
> > Submission5407 Authors

---

### Official Review · Reviewer_cmkV · 2023-11-01

**Soundness:** 3 good
**Presentation:** 3 good
**Contribution:** 3 good
**Rating:** 6
**Confidence:** 3

**Summary:**

Simplified spiking neuron models, such as LIF, have trouble with multi-scale sequence modeling, or learning tasks involving learning complex temporal dynamics. While others have developed methods to account for this, training speed and performance is still relatively subpar. In this paper, the authors utilize biological inspiration from hippocampus pyramidal neurons and their multi-compartmental modeling to design a new generalized multi-compartment model which allows for arbitrary numbers of compartments. They also introduce a parallel implementation of this model for faster training on GPU accelerated hardware while accounting for the reset mechanism, unlike previous works.

**Strengths:**

1) The new multi-compartment model performs better than other memory-enhanced models and single compartment models on sequence modeling tasks with relatively similar parameter counts on standard benchmarks.
2) The topic of better exploiting temporal dynamics during SNN training is important for their development.
3) The parallel implementation, especially accounting for reset, is an important and useful contribution as SNNs are generally slow to train and have yet to be parallelized effectively.

**Weaknesses:**

1) Regarding Figure 4: while the parallel model was compared to its serial implementation in terms of speed up, what is the ratio/training time difference between PMSN and PSN? If PSN is faster, since it is only a single compartment model, how significant is this difference? I am not referring to the computational cost but instead training acceleration only.
2) Is the parallel implementation/methodology that incorporates the reset mechanism general enough to be applied to single compartment models?

**Questions:**

Please see above.

---

> ### Author Response · Authors · 2023-11-15
> **Response to Reviewer cmkV**
>
> > **Weakness 1:** Regarding Figure 4: while the parallel model was compared to its serial implementation in terms of speed up, what is the ratio/training time difference between PMSN and PSN? If PSN is faster, since it is only a single compartment model, how significant is this difference? I am not referring to the computational cost but instead training acceleration only.
>
> **Response:** Thanks for your insightful comment. Following your suggestions, we have conducted a comparative analysis of the training speed between the PMSN and PSN models. This involves measuring each model's training time across various tasks with different time window lengths, with identical training configurations and network architectures for a fair and direct comparison. The results, including time elapsed (seconds/epoch) and classification accuracy (%), are as follows:
>
> | Dataset | S-MNIST (T=784) | Sequential CIFAR10 (T=32) | Sequential CIFAR10 (T=1024) |
> | ------- | --------------- | ------------------ |-------------------- |
> | Speed (PMSN) | 35.3235 | 61.7562 | 73.2060 |
> | Accuracy (PMSN) | 99.40% | 90.97% | 82.14% |
> | Speed (PSN) | 21.1776 |  34.0634 | 49.0240 |
> | Accuracy (PSN) | 97.90% | 88.45% | 55.24% |
>
> The results clearly show that the PMSN model, with more complex and sophisticated multi-compartment dynamics (five compartments in PMSN vs. one in PSN), trains slightly slower than the PSN model. However, the difference in training speed is not significant (with a ratio of less than 2 across all the experiments), demonstrating the efficiency of our parallelization approach. Furthermore, this minor trade-off in speed is outweighed by the superior sequential modeling capabilities of PMSN, particularly evident in its performance on the Sequential CIFAR-10 dataset. This underlines the value of the PMSN model in applications where enhanced temporal signal processing capacity is critical.
>
> &nbsp;
>
> > **Weakness 2:** Is the parallel implementation/methodology that incorporates the reset mechanism general enough to be applied to single compartment models?
>
> **Response:** We appreciate your insightful question. This reset mechanism can indeed be seamlessly transferred to the single-compartment Integrate-and-Fire (IF) model, thereby enabling parallelizable training for this model. The generalization is straightforward and involves a simple modification. Specifically, it requires replacing the input of Equation 13, denoted as $I_h[t]$, with the input of the single-compartment model. For instance, the replaced input can be denoted as $I^l[t]=\mathcal{W}^l S^{l-1}[t]$, which adheres to a common practice of single-compartment models regardless of the dynamics of synaptic decay. Consequently, Equations 13 and 14 effectively formulate a parallel IF model that incorporates our reset mechanism.
>
> &nbsp;
>
> We sincerely hope that our response has addressed your concerns. We will include these additional results in the revised manuscript and Appendix.

---

> > ### Author Response · Authors · 2023-11-23
> > **A Gentle Reminder for Final Hours for Discussion**
> >
> > Dear reviewer cmkV,
> >
> > &nbsp;
> >
> > As we approach the final hours of the discussion period, we take this opportunity to respectfully inquire if there are any remaining questions or concerns about our manuscript. As we discussed above, the training speed gap is not significant, and the reset mechanism is able to be effortlessly generalized. Your insights are greatly valued, and we deeply appreciate the time and effort you have dedicated to reviewing our work.
> >
> > &nbsp;
> >
> > Sincerely,
> >
> > Submission5407 Authors

---

### Author Response · Authors · 2023-11-23
**Summarize Contributions of the Paper**

Dear reviewers and AC,

&nbsp;

Thank you all for your invaluable time and effort in reviewing our paper. Your constructive suggestions have been instrumental in enhancing the quality of our manuscript. We would like to highlight the main contributions of our work as follows:

* We've focused on bridging the performance gap between biological neurons and SNNs in processing sequential signals, exploring the temporal dynamics of spiking neuron models, a significant challenge that has not been sufficiently explored to date [supported by reviewer cmkV, QBv1].

* Drawing from the multi-compartment model developed in computational neuroscience, we've adapted it for deep SNNs and real-world pattern recognition tasks. The multi-compartment model is largely unexplored in SNNs, and our model is the first to incorporate more than two compartments  [supported by reviewer 9iDX].

* Addressing efficiency challenges of the multi-compartment model, we've proposed a parallel multi-compartment model that enables GPU acceleration in the temporal dimension, significantly enhancing training speed, which is a significant and useful contribution [supported by reviewer cmkV, QBv1, Chwq].

* Our model addresses the non-linear recurrent problem, incorporating the reset mechanism into neural dynamics, which was previously omitted in parallel SNN works. Experimental results demonstrate its positive impact on performance [supported by reviewer cmkV, kRbB].

* The PMSN model shows considerable performance enhancement in sequential tasks [supported by reviewer cmkV, 9iDX, kRbB, Chwq]. What's more, it strikes a fine balance between capturing intricate temporal dynamics and maintaining computational efficiency [supported by reviewer cmkV, QBv1, 9iDX].

Following your feedback, we have made further improvements to our manuscript and appendix, highlighted in blue for your convenience. Once again, we are immensely grateful for your thorough review and valuable suggestions.

&nbsp;

Sincerely,

Submission5407 Authors

---

### Meta-Review · Area_Chair_KB1t · 2023-12-05

**Metareview:**

The paper proposes a new neuron model for brain-inspired Spiking Neural Networks, as well as a parallel implementation strategy, with the goal of improving performance on tasks which involve complicated temporal characteristics. The proposed multi-compartment model is evaluated across multiple tasks, and the authors report improved performance.

Reviewers generally agreed that the performance comparisons were sound, but there was significant discussion  what precisely the goal of the paper is. The authors, in several places, argued that they are not concerned with biological realism (but the focus is just neuromorphic engineering, i.e. building well-performance spiking neuron models), in others they explicitly state that their goal is to capture the rich dynamics of biological neurons.

One reviewer asked for additional analyses, which the authors performed, but then finally did not comment on the outcome of these analyses, and not update their score. However, the additional analyses were read by the AC, and considered in making the final decision.

However, some questions about the general applicability and impact of this approach remained. It seems clear that there is material for an interesting and sound paper-- however, as it stands, the submitted paper (even in its revised form) would benefit from further clarification. Therefore, the final decision was not to accept the paper.

**Justification For Why Not Higher Score:**

see above

**Justification For Why Not Lower Score:**

see above

---

### Decision · Program_Chairs · 2024-01-16

Reject